# FSL-MIC: An Attentional Few-Shot Learning Framework for EEG Motor Imagery Classification

## Abstract

Electroencephalography (EEG) is a key non-invasive technique used to investigate brain activity, particularly in motor imagery (MI) research. Traditional methods for classifying EEG signals often rely on handcrafted features and heuristic parameters, which can limit generalization across tasks and subjects. Recent advances in deep learning, particularly few-shot learning (FSL), offer promising alternatives to improve classification accuracy in scenarios with limited training data. This study explores the effectiveness of FSL algorithms, including Relation Networks, to enhance MI classification. It also examines how transfer learning and data augmentation techniques contribute to improving classification performance.

We propose a novel framework with three core modules—feature embedding, attention, and relation—that facilitates the classification of unseen subject categories using only a few labeled samples. The attention mechanism identifies key features related to the query data, while the relation module predicts query labels by modeling relationships between support and query data across subjects. Our experimental results demonstrate the effectiveness of our approach on two benchmark datasets, BCI 2a and BCI 2b, as well as our experimental dataset. The proposed FSL framework significantly outperforms traditional methods, offering promising applications in real-time Brain-Computer Interface (BCI) systems across various EEG setups. This research advances the understanding of machine learning in EEG applications and highlights the potential of FSL techniques in overcoming the challenges of limited training data in MI classification.

## 1 Introduction

Electroencephalography (EEG) is a non-invasive, cost-effective, and portable technique for measuring the brain's electrical activity Lashgari et al. (2020). With its high temporal resolution, EEG can capture rapid changes in brain activity, making it a valuable tool for studying dynamic brain processes such as motor imagery (MI). MI, defined as the mental simulation of motor actions, generates distinct EEG patterns and is central to Brain-Computer Interface (BCI) research. Motor imagery-based BCIs have been applied in various domains, including rehabilitation, communication, and assistive device control. These systems typically capture EEG signals during MI tasks, extract relevant features, and classify them to interpret the user's intended actions Lashgari et al. (2020; 2021).

Traditionally, feature extraction and classification of EEG signals for MI tasks have been performed in the time, frequency, and spatial domains. Many of these methods rely on handcrafted features, which are based on human expertise Altaheri et al. (2023); Lashgari et al. (2020). For example, time-frequency analysis techniques such as short-time Fourier transform or wavelet transforms are commonly used, while filter-bank common spatial patterns (FBCSP) have shown efficacy in the spatial domain Blanco-Díaz et al. (2024). However, FBCSP uses fixed temporal windows, which may neglect individual variability and fail to leverage the full potential of time-domain information. Furthermore, handcrafted features often require heuristic parameter settings, such as predefined frequency bands, which can limit generalizability and reduce classification accuracy across different tasks and subjects.

To overcome these limitations, recent research has focused on using deep learning for automatic feature extraction and classification in EEG MI tasks Lashgari et al. (2021); Kumari et al. (2024). Neural networks can derive relevant features directly from raw EEG data, eliminating the need for manual feature engineering and improving classification performance. However, acquiring large, high-quality training datasets for MI classification remains a challenge. This is due to the sophisticated EEG equipment required, the need for controlled, noise-free environments, and the demands placed on participants, which often lead to fatigue and attrition. Moreover, deep learning models typically require large amounts of data to achieve optimal performance, making MI classification with limited data especially challenging.

To address these issues, Lashgari et al. (2020) demonstrated that data augmentation (DA) techniques can enhance the classification accuracy of EEG signals. Their study applied various DA methods, including random rotation, translation, scaling, and flipping, to improve model robustness and generalization. Building on this, Lashgari et al. (2021) introduced a convolutional neural network (CNN) with an attention mechanism and DA strategies, achieving superior classification accuracy on benchmark MI datasets (BCI Competition IV 2a and 2b). Their end-to-end framework provided automatic feature extraction from raw EEG data and interpretable outcomes, surpassing existing methods.

While data augmentation can improve performance, it may not fully resolve the challenge of limited data in MI classification. In this context, few-shot learning (FSL) has emerged as a promising alternative, allowing models to generalize from minimal training data An et al. (2020); Ahuja & Sethia (2024). FSL focuses on learning from few labeled examples, typically involving training on a broader set of tasks, with each task having a limited number of labeled instances. During testing, the model must adapt to new tasks using only a handful of examples. Several studies have highlighted the potential of FSL in motor imagery classification, emphasizing the need for further exploration across various datasets and conditions.

This paper investigates the efficacy of few-shot learning in the context of motor imagery classification, comparing it with data augmentation techniques. We introduce a novel framework, termed FSL-MIC (Few-Shot Learning for Motor Imagery Classification), which minimizes the need for extensive recalibration and allows users to retrain the model with minimal exercises. The proposed FSL-MIC architecture leverages domain knowledge to require significantly less training data than traditional methods while effectively decoding users' hand movements.

Our goal is to enable accurate classification of unseen subject categories using limited labeled samples, with paired support and query data from other subjects. The proposed method consists of three interconnected modules: feature embedding, attention, and relation. The feature embedding module extracts relevant features from the support and query data, the attention module focuses on key features in the support data relevant to the query, and the relation module predicts the label of the query sample based on the embedding space. By modeling relationships between paired samples across subjects, our approach aims to generalize well to classify query signals from unseen subjects, even with limited labeled examples.

The contributions of this paper are as follows:

- Introducing a class of architectures for EEG meta-learning that enables rapid adaptation based on minimal training observations.

- Presenting the FSL-MIC framework, which combines temporal convolutions and attention mechanisms to reduce training time and address variability in EEG characteristics.

The paper is organized as follows: Section II reviews relevant literature. Section III describes the dataset and preprocessing steps used in developing the FSL-MIC framework. Section IV details the proposed architecture. Section V presents experimental results and evaluation scenarios. Finally, Section VI concludes the paper.

## 2  RELATED WORKS

Recent advancements in deep learning (DL) have significantly enhanced automatic feature extraction and classification in various domains, including EEG signal analysis Lashgari et al. (2020); Sharma & Meena (2024). Consequently, research efforts have focused on neural network (NN) architectures,

training procedures, regularization techniques, optimization methods, and hyperparameter tuning, often resulting in substantial improvements in decoding accuracy Sharma & Meena (2024).

Despite their potential, training deep neural network architectures from scratch poses challenges, primarily due to the substantial amount of training data required to achieve high classification accuracy. This challenge is particularly pertinent for motor imagery (MI) classification, where acquiring a large volume of high-quality training samples is difficult Lashgari et al. (2020). Gathering such data necessitates expert training, sophisticated EEG equipment, and a controlled, noise-free environment. Additionally, MI tasks can be time-consuming and mentally taxing for participants, who must minimize movements and maintain intense concentration. As a result, data collection often requires multiple sessions, which can lead to participant attrition over time. Traditional methods relying on large datasets thus face difficulties in generalizing across subjects and tasks.

To mitigate these limitations, researchers have turned to advanced architectures that leverage data augmentation (DA) techniques. In 2021, Lashgari et al. proposed an end-to-end convolutional neural network (CNN) incorporating an attention mechanism and various DA techniques Lashgari et al. (2021). Their model was tested on two benchmark MI datasets, the Brain-Computer Interface (BCI) Competition IV 2a and 2b. Additionally, they developed a high-density EEG dataset that included both MI and motor execution (ME) tasks, significantly contributing to the field.

However, while DA improves model robustness, it may not fully address the challenge of limited training samples per subject. In this context, few-shot learning (FSL) has emerged as a promising solution Fei-Fei et al. (2006). This approach allows models to generalize from only a few labeled examples by learning patterns from other subjects An et al. (2020). For EEG-based MI classification, this is particularly useful as it reduces the dependency on extensive datasets while also addressing the variability between participants. In 2023 An et al. evaluated their proposed method with three different embedding modules on cross-subject and crossdataset classification tasks using brain–computer interface (BCI) competition IV 2a, 2b, and GIST datasets An et al. (2023).

Although few-shot learning methods for EEG-based MI classification remain sparse in the literature, some studies have begun to explore its potential. For instance, one study utilized one-shot learning with intracranial electroencephalography (iEEG) signals, employing local binary patterns to extract features and learning prototype vectors in a hyper-dimensional space Burrello et al. (2019). Classification was performed by measuring distances between these prototype vectors and those from query signals. While this method showed encouraging results, it did not provide an end-to-end learning framework.

In another study, researchers proposed a deep neural network with a triplet loss function for classifying various time series data, including ECG recordings Narwariya et al. (2020). This approach highlighted the viability of few-shot learning for signal analysis, although it was limited to learning feature embedding networks that categorized embedded features using a fixed nearest neighbor classifier. A more recent contribution introduced a relation-based few-shot learning method that learns non-linear comparators by capturing end-to-end relationships between support data distributions and query signals, without relying on distance-based calculations between embedding vectors. This network architecture bore similarities to those used in image classification Sung et al. (2018), but it integrated a modified hierarchical spatial convolutional neural network (HS-CNN) suitable for 1D signal analysis and incorporated an attention module to emphasize critical signals in the support set.

In this paper, we propose a novel relation-based few-shot learning method that learns end-to-end relationships between support data and query signals, eliminating traditional distance-based calculations. Our network architecture builds upon recent developments in MI classification Lashgari et al. (2021). We evaluate this framework on two benchmark datasets (BCI Competition IV 2a and 2b) as well as a newly collected dataset, aiming to further validate the efficacy of few-shot learning for EEG-based MI tasks.

In summary, while few-shot learning for EEG-based MI classification is still in its early stages, the existing literature indicates that this approach holds significant promise for advancing research in this domain. It offers a powerful alternative to traditional methods, particularly in scenarios with limited data and high inter-subject variability.

# 3 METHOD

The proposed 2-way K-shot learning framework is illustrated in Fig. 1. This framework consists of three key modules: an embedding module, an attention module, and a relation module. In the following sections, we will explain each of these modules in detail.

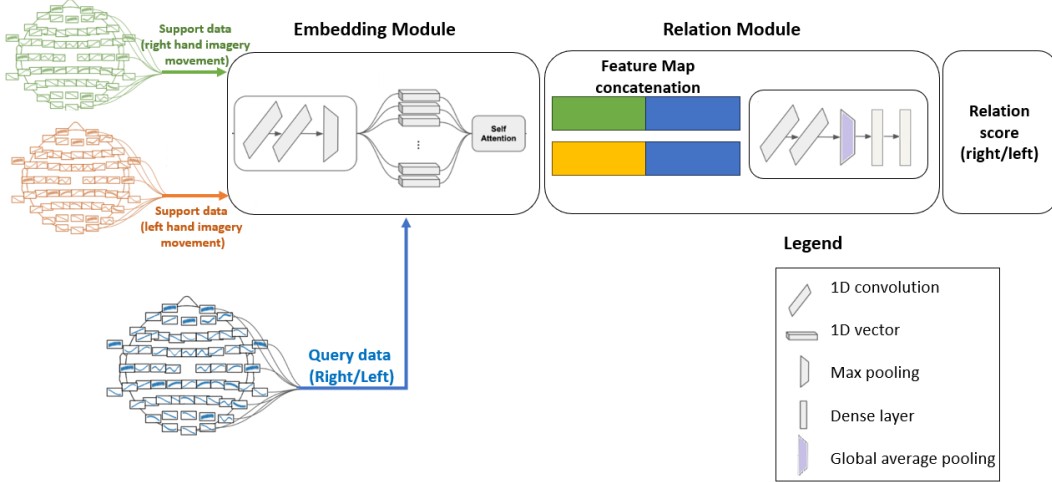

Figure 1: Overview of the proposed few-shot relation network. The framework includes two key modules: embedding via attention and a relation module. Given support and query data, we obtain feature vectors concatenated at several stages in our pipeline through an embedding module. These features are fed to an attention module that focuses on key features related to the query data based on attention scores. The attended features are further processed by a relation module that predicts relation scores from representative vectors and query features, ultimately assigning a label to the given query sample. Based on these predictions, we update the weights of the entire framework in a single training episode.

## 3.1 EMBEDDING MODULE

The embedding module extracts semantic features from the input EEG signals to facilitate classification. Convolutional models have been successful in many signal processing applications, as they allow temporally related inputs to be processed together via a sliding-window approach. This results in shared weights, where the same weight kernel is applied across the temporal domain (for a 1D convolutional model over time).

In our architecture (Figure 1), the convolutional layers reduce the number of parameters needed, while enabling the signal to maintain its spatial relations—both across time within each electrode and across electrodes on the head. The signal from each electrode channel is processed through the same convolutional base, producing an output matrix of dimension $C \times E$, where $C$ is the number of electrodes (or channels) and $E$ is the size of the embedding dimension. Consequently, the convolutional layers effectively reduce the input dimension to $E$, which is 100 times smaller than the original samples.

## 3.2 SELF-ATTENTION MECHANISM

In the self-attention part of the network, we first initialize the weights for the Query (Q), Key (K), and Value (V) matrices Vaswani (2017); Lashgari et al. (2020). The magnitudes of Q, K, and V are derived from the input (I) and the corresponding weights. The attention score $S$ is calculated as follows:

$$S = QK^T$$

Here, $S$ has a shape of $C \times C$, where $C$ is the number of channels. We then apply the Softmax function to $S$ to obtain the attention weights $W$:

$$W = \text{Softmax}(S)$$

Next, we compute the weighted values $M$:

$$M = WV$$

Each input's value for $M$ is then transformed using the hyperbolic tangent function to produce the final attention output $O$:

$$O = \tanh(M)$$

This attention layer is integrated after the convolutional base (see Figure 1), allowing each electrode channel to interact with every other channel, resulting in a matrix of scalar values. Summing across rows and normalizing these scalars produces a vector of attention scores. These scores are subsequently utilized to create a linear combination of all the electrode channel vectors, which is passed to the fully connected layers for classification.

A key feature of our model is its interpretability, which is enhanced by the attention mechanism. The attention scores for each EEG electrode channel can be visualized, allowing us to identify which electrodes are most influential in the model's predictions at different time points. While the full results, including data from multiple subjects, will be presented in our next paper, we have included a representative example from a single subject in this work to illustrate the model's interpretability.

Specifically, in the Supplementary Material, we provide a heatmap of attention scores across various electrode channels, where color intensity represents the importance of each electrode for the model's decision-making process. Brighter colors correspond to higher attention, highlighting the electrodes that contribute most to the classification task at each time step. This visualization offers valuable insights into the spatial distribution of attention across electrodes and demonstrates how the model prioritizes task-relevant features, such as specific brainwave patterns, over time.

By focusing on a single subject for this demonstration, we aim to provide an initial insight into the attention mechanism's interpretability, while recognizing that broader results involving multiple subjects will be presented in subsequent work.

While techniques like Grad-CAM (Gradient-weighted Class Activation Mapping) can visualize the model's focus on specific input regions, we prioritize the attention mechanism to improve prediction accuracy rather than focusing solely on interpretability Selvaraju et al. (2017); Li et al. (2020). This allows the CNN, in conjunction with the attention mechanism, to better classify EEG signals by emphasizing critical, task-relevant features across different time steps via sliding windows. Ultimately, the attention mechanism both enhances the model's prediction performance and provides a clear, interpretable view of which features are being prioritized in the decision-making process.

The embedding module takes pairs of $k$ support and query data as input, extracting semantic features for classification using convolutional layers. The self-attention mechanism further allows the model to weigh the importance of different input elements, which is particularly beneficial when the relationships between inputs, such as features from different electrodes, are crucial for model performance.

## 3.3 RELATION MODULE

In few-shot learning, the relation module is essential for comparing support samples with query samples to generate predictions. Rather than relying on traditional distance metrics, our approach utilizes a combination of convolutional neural networks (CNNs) and fully connected (FC) layers to assess the relationships between the embeddings of support and query samples. The CNN extracts rich feature representations, which are then processed through FC layers to learn non-linear relationships. This enables our model to effectively identify relevant support samples for each query without fixed distance calculations. By emphasizing learned relationships, our relation module enhances the

model's ability to generalize from limited examples, thereby improving classification performance in scenarios with scarce labeled data.

The relation module computes relation scores based on the class-representative vectors and query features. The class corresponding to the highest relation score is assigned as the predicted label for the query. During training, the network is optimized in an end-to-end manner using pairs of support sets and query signals from subjects in the training data. For testing, the label of a query signal is predicted using $k$ labeled support signals from an unseen subject. Although the illustration depicts a 2-way K-shot learning scenario, our proposed method can be easily extended to N-way K-shot learning configurations.

To estimate the label from the concatenated features of each support and query feature in the channel direction, we employ two convolutional layers with kernel sizes of $30 \times 1$ and $15 \times 1$, followed by a global average pooling layer and two fully connected layers with dimensions 256 and 100 (Figure 1). For training, we utilize a focal loss function defined as follows:

$$ \text{Loss} = -\sum_i \alpha(1 - p_i)^\gamma y_i \log(p_i) $$

where $y_i$ is 1 if the class of the supporting data matches that of the query data, and 0 otherwise. The parameters $\alpha$ and $\gamma$ control the balance between classes and the focusing effect, respectively.

We optimize the model parameters using the Adam optimizer, with a batch size set to 164 and an initial learning rate of $10^{-4}$. The learning rate decays exponentially at a rate of 0.033% with each iteration, and the model is saved when the validation loss reaches its minimum.

## 4 EXPERIMENTAL RESULTS

### 4.1 DATASET

We used three datasets in this study: (1) A dataset that we collected ourselves, (2) the BCI 2a dataset, and (3) the BCI 2b dataset (Figure 5).

#### 4.1.1 OUR DATASET

Seven healthy volunteers (3 male and 4 female), all right-handed and aged between 23 and 30 years (mean age: 28), participated in the study, providing written informed consent prior to the experiment. Participants were seated in a chair positioned 80 cm from an LCD screen, with both hands resting on a table while holding a tennis ball in each hand. They were instructed to remain relaxed and minimize movement and eye blinks throughout the sessions. When required to respond, participants were to squeeze the tennis ball while avoiding any tension in their arms or shoulders.

Each experimental session included both motor execution (ME) and motor imagery (MI) tasks (see Figure 2) and was repeated twice, resulting in a total of four sessions; however, this study focuses exclusively on the MI sessions. Each session lasted between 30 to 40 minutes, with breaks of 10 to 15 minutes in between to help reduce fatigue. The entire experiment, including setup, was designed to be completed in under 3 hours. EEG data were recorded at a sampling rate of 250 Hz using 64 active electrodes (BrainVision actiCHamp) arranged according to the 10/20 montage. Additionally, bipolar electromyography (EMG) electrodes were placed on the brachioradialis of both arms to monitor any movement during the MI tasks.

During the MI sessions, participants were instructed to imagine the repetitive hand movements outlined in session 1 while fixating on a cross displayed on the screen. A total of 100 trials for both left- and right-hand imagery were collected in each session, with all other task aspects mirroring those of session 1. This setup allowed for the assessment of motor-related EEG oscillations and ensured at least minimal voluntary control over these oscillations. In total, we collected 200 trials of ME and 200 trials of MI for each participant. The underlying data for this study have been uploaded to Figshare and are available at the following link: [xxx].

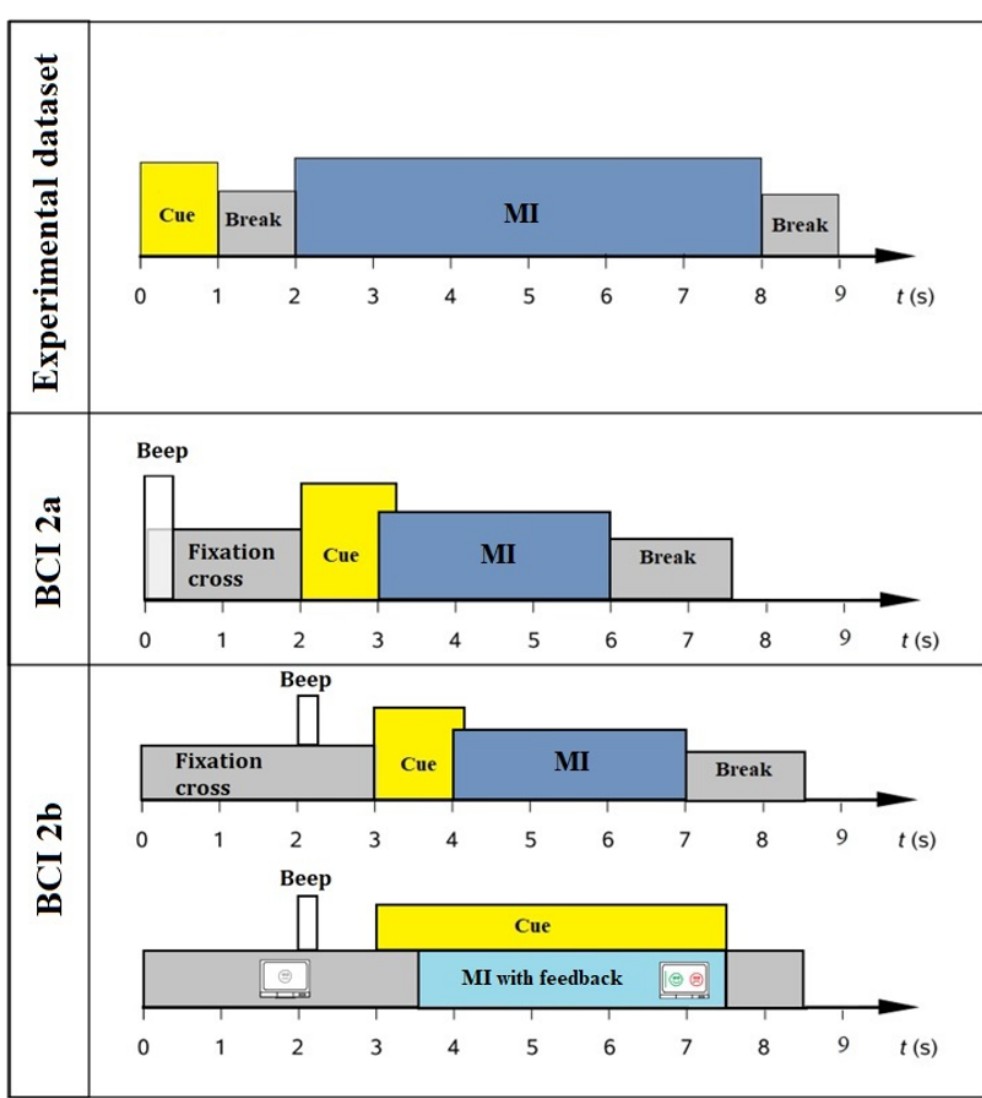

Figure 2: The experimental paradigms for our experimental dataset, BCI 2a, and BCI 2b..

### 4.1.2 BCI 2A

The BCI 2a dataset contains EEG data from 9 healthy participants, 2 sessions per participant. Each session is made up of 288 trials, resulting in 5184 trials overall. No feedback was provided. Twenty-two Ag/AGCL channels were used to record EEG. The signals were sampled with 250 Hz and bandpass filtered between 0.5-100 Hz. To compare our results with previous studies, we focused on the C3, CZ, and C4 electrodes (see Figure 2).

### 4.1.3 BCI 2B

The BCI 2b dataset contains EEG data from another 9 healthy participants. For each participant, 5 sessions of data were collected. Each of the first 2 sessions has 120 trials, and each of the last 3 sessions has 160 trials. The total number of trials is thus 6480. Two types of trials are included in these datasets: left- and right-hand MI. The first 2 sessions contain training data without feedback, while the last three sessions gave a smiley face as feedback. The EEG data is again collected over the C3, CZ, and C4 electrodes, which were placed following the international 10–20 system. The sampling frequency was 250 Hz. Figure 2 visualizes the summary of the three datasets.

## 4.2 EXPERIMENTAL SETTING

The performance of various CNN-based attention models was evaluated across three distinct datasets: BCI 2a, BCI 2b, and an experimental dataset. Each dataset comprises different channel configurations—22 channels for BCI 2a, 3 channels for BCI 2b with feedback, and 64 channels for the experimental dataset. The evaluation metrics include standard accuracy and DA accuracy, reported with means and standard deviations.

For our experiments, we employed 9-fold and 7-fold cross-validation, training the model with data samples from 8 or 7 subjects and testing on the remaining subjects for each validation episode. Each training set included all experiments from a subject except the last one, which served as the validation set. The test set comprised the complete dataset of the remaining subjects. During training, support and query data samples were randomly selected from both the training and validation sets at each iteration. For testing, we divided the samples from each class into two groups: 20 samples designated as support and the rest as query data.

To evaluate the performance of our proposed model (RelationNet-attention), we assessed the classification accuracy for K-shot classification models with K = $\{1, 5, 10, 20\}$. For K = $\{1, 5, 10, 20\}$ experiments, K support samples were randomly chosen from the 20 support samples.

Additionally, we compared our method against two supervised learning models: (1) CNN-attention-Few, trained with only 40 samples (20 for left and 20 for right) in the support set of the testing subjects, and (2) CNN-attention-All, trained with all training samples from all subjects except one. Each experiment was repeated 10 times, and the average accuracy was calculated for all cases, accounting for potential variations in support sets.

## 4.3 RESULTS AND ANALYSIS

### 4.3.1 BCI 2A DATASET

Table 1 presents the results of our study. The BCI 2a dataset results show that the CNN-attention-All model outperformed other variants, achieving the highest accuracy of 89.1% ± 0.4 and DA accuracy of 93.6% ± 0.3. This result underscores the robustness of the model when leveraging all data channels. In comparison, the CNN-attention-Few model, which utilizes fewer shots, reported a significantly lower accuracy of 62.8% ± 4.3 and DA accuracy of 66.3% ± 3.3, indicating the importance of using more data channels for achieving higher classification performance.

Notably, BCI 2a does not include neurofeedback during recording, which makes the classification task more challenging despite having 22 channels. The absence of feedback requires the model to rely more heavily on learning complex patterns from the raw signals, making the classification process more difficult.

The RelationNet-attention model demonstrated a clear trend of performance improvement as the number of shots increased. Starting from 1 shot, the accuracy was 63.1% ± 4.1, improving to 72.6% ± 2.9 with 20 shots. Similarly, the DA accuracy increased from 67.9% ± 5.3 to 77.8% ± 2.4. These results suggest that increasing the shot count enhances the model's learning capacity, resulting in better generalization across unseen data.

### 4.3.2 BCI 2B DATASET

For the BCI 2b dataset, which includes a neurofeedback mechanism, the results followed a similar trend. The CNN-attention-All model achieved the highest performance, with an accuracy of 86.28% ± 0.8 and DA accuracy of 87.83% ± 0.7. This result emphasizes the effectiveness of using all available channels in this particular dataset with feedback loops. While BCI 2b contains only 3 channels, the inclusion of neurofeedback simplifies the classification process, compensating for the reduced number of channels by providing real-time cues that enhance model training and performance.

The CNN-attention-Few model, on the other hand, performed less effectively, with an accuracy of 71.3% ± 6.1 and DA accuracy of 72.77% ± 5.2, although it still benefits from the feedback mechanism.

The RelationNet-attention model again exhibited improved performance with an increasing number of shots. Accuracy improved from 59.9% ± 7.2 with 1 shot to 73.2% ± 6.1 with 20 shots, while DA accuracy increased from 62.9% ± 4.2 to 75.9% ± 5.7. These results suggest that the model's ability to adapt to varying shot counts is beneficial for datasets with limited channels and feedback, contributing to its flexibility across different settings.

### 4.3.3 EXPERIMENTAL DATASET

For the experimental dataset, which employs 64 channels, the CNN-attention-All model again achieved the highest accuracy at 81.24% ± 1.1 and a DA accuracy of 83.42% ± 2.1. This demonstrates the scalability of the model across datasets with more data channels. The CNN-attention-Few model, while still showing reasonable performance, recorded lower results with an accuracy of 69.2% ± 4.3 and DA accuracy of 74.11% ± 5.3, illustrating the challenges of using fewer shots in datasets with higher channel counts.

The RelationNet-attention model continued to show improvements with increased shots. Accuracy rose from 54.9% ± 5.9 with 1 shot to 68.2% ± 5.1 with 20 shots. Similarly, DA accuracy improved from 60.1% ± 6.3 to 68.9% ± 3.2, underscoring the benefits of increasing the number of training samples for better model generalization.

To enhance visualization, we present the model performance on the BCI 2a, BCI 2b, and experimental datasets in Figure 3.

Table 1: Performance of different models on BCI 2a, BCI 2b, and Experimental datasets.

| Dataset | Model | Accuracy ± STD (%) | DA Accuracy ± STD (%) |
|---|---|---|---|
| BCI 2a | CNN-attention-All | 89.1 ± 0.4 | 93.6 ± 0.3 |
| | CNN-attention-Few | 62.8 ± 4.3 | 66.3 ± 3.3 |
| | RelationNet-attention (1 shot) | 63.1 ± 4.1 | 67.9 ± 5.3 |
| | RelationNet-attention (5 shots) | 69.4 ± 3.3 | 73.8 ± 3.1 |
| | RelationNet-attention (10 shots) | 70.8 ± 2.7 | 75.3 ± 2.5 |
| | RelationNet-attention (20 shots) | 72.6 ± 2.9 | 77.8 ± 2.4 |
| BCI 2b | CNN-attention-All | 86.28 ± 0.8 | 87.83 ± 0.7 |
| | CNN-attention-Few | 71.3 ± 6.1 | 72.77 ± 5.2 |
| | RelationNet-attention (1 shot) | 59.9 ± 7.2 | 62.9 ± 4.2 |
| | RelationNet-attention (5 shots) | 68.2 ± 5.8 | 72.4 ± 4.6 |
| | RelationNet-attention (10 shots) | 71.5 ± 4.2 | 74.9 ± 3.9 |
| | RelationNet-attention (20 shots) | 73.2 ± 6.1 | 75.9 ± 5.7 |
| Experimental | CNN-attention-All | 81.24 ± 1.1 | 83.42 ± 2.1 |
| | CNN-attention-Few | 69.2 ± 4.3 | 74.11 ± 5.3 |
| | RelationNet-attention (1 shot) | 54.9 ± 5.9 | 60.1 ± 6.3 |
| | RelationNet-attention (5 shots) | 63.4 ± 5.2 | 67.1 ± 4.8 |
| | RelationNet-attention (10 shots) | 65.7 ± 4.1 | 68.4 ± 3.9 |
| | RelationNet-attention (20 shots) | 68.2 ± 5.1 | 68.9 ± 3.2 |

### 4.4 CONCLUSION

In this study, we propose a two-way few-shot classification network designed to effectively classify EEG-based motor imagery (MI) data from unseen subjects. Our results indicate that, although BCI 2b is utilized in the referenced study An et al. (2020), our model outperforms it, achieving superior accuracy across all three datasets. Specifically, the CNN-attention-All model consistently demonstrated enhanced performance in terms of both overall accuracy and DA accuracy. The presence of neurofeedback in BCI 2b simplified the classification process, allowing our model to achieve high accuracy with only three channels, despite the limited number of input channels. Conversely, the absence of neurofeedback in BCI 2a increased classification difficulty, even with 22 channels available, highlighting the additional challenge of learning from raw signals without real-time guidance. Our approach differs from that of An et al. (2023) in two key ways: we utilize dataset-specific data augmentation (DA) techniques based on the findings of Lashgari et al., rather than focusing on fine-tuning (FT), and we employ a tailored dataset split strategy that better aligns with the character-

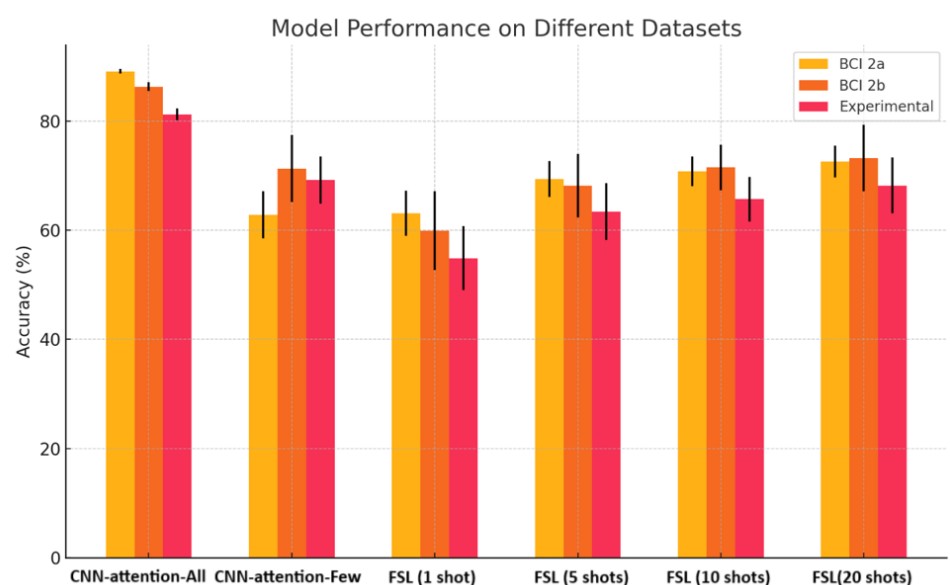

Figure 3: Performance of different models on the BCI 2a, BCI 2b, and experimental datasets. The bars represent the accuracy achieved by each model, with each trial corresponding to a different experimental condition, including CNN-attention-all, CNN-attention-few, and various few-shot learning scenarios (1-shot, 5-shot, 10-shot, and 20-shot). The error bars indicate the standard deviation, providing insight into the variability of performance across these trials.

istics of our data and enhances the robustness of our results, particularly for online BCI applications. These differences, combined with our integration of few-shot learning, offer a more practical and scalable solution for BCI systems with limited labeled data.

Our experiments have demonstrated that few-shot learning with limited samples yields promising results in classifying EEG data, significantly reducing the time and effort required for BCI training. The FSL-MIC framework leverages this paradigm, requiring fewer training examples per class, which minimizes data preparation time and computational load, leading to faster training cycles. Additionally, techniques like RelationNet help the model generalize efficiently from limited data, achieving optimal performance more quickly. This is particularly beneficial for patients with disabilities, who often face fatigue and challenges during extended training sessions. By alleviating these burdens, we can make BCI systems more accessible and user-friendly for individuals with motor impairments.

Furthermore, our model's architecture is not restricted to EEG data alone; it can be adapted to other types of time-series data exhibiting substantial inter-subject variability. This versatility expands the potential applications of the proposed framework beyond MI classification, making it a viable solution for time-series classification across a wide range of domains, including healthcare, finance, and autonomous systems. Notably, the model also shows promise for tasks such as hand gesture classification, enhancing its applicability in human-computer interaction scenarios.

Overall, the simplicity and effectiveness of this few-shot learning approach, combined with the attention mechanism, positions it as a promising solution for EEG-based MI classification. The method offers a balanced trade-off between performance and computational efficiency, providing a practical tool for real-time BCI systems and enhancing the overall user experience. Additionally, we have incorporated some of our architectural settings similar to those in the referenced paper for a comprehensive comparison of our proposed framework.

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
