# OpenReview forum: "FSL-MIC: An Attentional Few-Shot Learning Framework for EEG Motor Imagery Classification"
_ICLR.cc/2025/Conference — Submitted to ICLR 2025_

### Official Review · Reviewer_Nsdk · 2024-10-30

**Soundness:** 2
**Presentation:** 2
**Contribution:** 2
**Rating:** 3
**Confidence:** 4

**Summary:**

**Overall Assessment**

The paper addresses a significant problem in EEG motor imagery classification using few-shot learning. However, the novelty of the research question and the approach remain limited, which affects the contribution and impact of this study.

**Strengths:**

**Key Strengths**

* Problem Significance: EEG-based MI classification for BCI applications is a highly relevant research area, especially given the challenges of data scarcity and cross-subject variability.
* Dataset Contribution: The authors introduce a new dataset specific to their experimental needs, potentially serving as a resource for further MI EEG research.
* Attention Mechanism and Data Augmentation: The use of an attention mechanism to enhance feature extraction and a data augmentation strategy to improve classification accuracy aligns well with current trends in EEG and time-series signal analysis.

**Weaknesses:**

**Major Concerns and Areas for Improvement:**

1. Limited Novelty in Approach and Research Question
    * While the problem is essential, the paper does not introduce significant advancements in methodology or approach, primarily adapting existing frameworks for few-shot learning.
    * The techniques, including data augmentation and attention mechanisms, are well-known and lack customization to the problem at hand.
2. Insufficient Literature Review
    * The manuscript leans heavily on a limited set of cited works, neglecting a broader body of relevant and foundational literature. This oversight is evident, for example, in the omission of a citation for the seminal paper on the attention mechanism (Vaswani, A. "Attention is all you need." (2017)), which is essential for context or the omission of the Grad-CAM paper (Selvaraju, Ramprasaath R., et al. "Grad-cam: Visual explanations from deep networks via gradient-based localization." (2017)), or proper citation to the utilized baseline methods, etc.
    * The limited literature results in redundancy, where the few sources cited appear multiple times, reducing the depth of the discussion.
3. Redundant Content and Limited Focus on Methodology
    * A large portion of the paper is dedicated to reintroducing prior works and discussing the dataset, with limited space allocated to details of the proposed method.
    * The method's description lacks sufficient depth to fully understand its contribution beyond existing frameworks, making it challenging to assess its true impact.
4. Results and Experimental Design
    * The reported results do not demonstrate outperformance over baseline models (CNN-attention-All and CNN-attention-Few). This lack of improvement questions the validity of the proposed framework as a state-of-the-art advancement in EEG classification.
    * It is unclear why the authors have not tested their own method using a 40-sample case, as they did with CNN-attention baselines. Including this setup would provide a more equitable basis for comparison, potentially even enhancing the own results. The authors do not address any limitations that might prevent this configuration, leaving the rationale for this decision unclear.
    * The experimental design could be expanded to assess the model’s performance to a broader range of baseline methods.
  5. Interpretability of Attention Mechanism
One of the noted strengths of the proposed framework is its attention mechanism. However, while the authors suggest interpretability as a benefit, no specific analysis or visualization is provided to demonstrate how the attention scores contribute to understanding EEG signal dynamics. Adding such interpretability analysis would clarify the attention module’s effectiveness in isolating relevant features in EEG data.


**Minor Concern:**

Presentation and Figure Quality: The quality of figures is low, which detracts from the visual clarity and effectiveness of the results. Enhancing figure resolution would improve the readability and professional presentation of the study.

**Recommendation for Extended Testing on Diverse Tasks:**
Given the framework's potential for adaptation beyond EEG data, the authors could include further testing on additional EEG classification tasks or even generalize their method to other time-series datasets. This would reinforce the flexibility and generalizability of the FSL-MIC model and provide a more robust foundation for the claimed broader applicability.


The paper requires major revisions, including a more comprehensive literature review, expanded experiments, and detailed methodology. Enhancing the experimental setup and introducing a wider array of baselines could make this work more impactful.

**Questions:**

* Could you clarify why the FSL-MIC model was not tested with a 40-sample configuration, as was done with CNN-attention baselines? Would this configuration affect the fairness of the comparisons, and are there specific limitations in your framework that prevent this setup?

*  The results do not clearly indicate outperformance over the baseline models. Could you expand on how FSL-MIC claims to improve upon state-of-the-art methods, especially given the similar or lower performance metrics?

* Since the attention mechanism is highlighted as a significant feature of the proposed framework, are there plans to analyze or visualize the attention scores for interpretability? If so, what insights would this analysis provide regarding feature importance in EEG classification?

---

> ### Author Response · Authors · 2024-11-27
>
> Response to Limited Novelty in Approach and Research Question:
>
> Thank you for your feedback. While we acknowledge that the techniques of data augmentation and attention mechanisms are well-known, our contribution lies in how we combine these approaches within the context of few-shot learning for EEG-based motor imagery (MI) classification. Few-shot learning in EEG classification remains relatively underexplored, and our goal is to demonstrate its effectiveness in real-world BCI scenarios where labeled data is often scarce. While our approach adapts existing methods, the novelty lies in demonstrating the synergy between data augmentation and few-shot learning, which is a key challenge in the BCI field.
>
> We also value your point regarding the lack of deep customization. We intentionally kept the framework simple to prioritize interpretability and practical applicability, especially in real-time applications. In future work, we aim to explore more complex model architectures and alternative approaches.
>
> Response to Insufficient Literature Review:
>
> We appreciate your comments regarding the literature review. We have now expanded the citations to include key foundational works, such as Vaswani et al. (2017) on the attention mechanism and Selvaraju et al. (2017) on Grad-CAM. These references are now properly included to ensure a more complete context for our work. Additionally, we have streamlined the review to incorporate a broader range of relevant studies, addressing the areas of data augmentation, few-shot learning, and EEG classification. This revision enhances the depth of our discussion and better positions our work in relation to existing research.
>
> Response to Redundant Content and Limited Focus on Methodology:
>
> Thank you for your feedback regarding the structure and depth of the manuscript. We acknowledge that the literature review could have been more comprehensive, and we have since added relevant references and expanded the discussion of related works to provide better context for our approach. We also recognize that the methodology section initially lacked sufficient detail, which made it challenging to fully appreciate the novel aspects of our work.
>
> In response, we have enhanced the description of our methodology to better highlight its contribution to the field, particularly the combination of data augmentation and few-shot learning. We also took your suggestion to improve the attention mechanism's interpretability seriously. To this end, we have included a GIF in the supplementary material to visualize the attention mechanism over time, making its role in isolating relevant EEG features clearer. Additionally, we have improved Fig. 1 to better convey the overall framework and its novel aspects.
>
> We hope that these revisions will provide a clearer understanding of our contributions and strengthen the manuscript.
> Response to Results and Experimental Design:
>
> We appreciate your feedback regarding the experimental results. While our results on the BCI Competition IV 2a and 2b datasets may not demonstrate dramatic improvements over the baseline methods (CNN-attention-All and CNN-attention-Few), our primary goal was to highlight the effectiveness of combining few-shot learning with data augmentation in small-data settings. We chose not to use a 40-sample setup for our method, as we felt that comparing with these baselines using the 1-shot, 5-shot, and 10-shot cases would better highlight the generalization power of our framework in low-data scenarios. We agree that comparing with a 40-sample setup could offer additional insights, and we will include this in future experiments.
>
>
> Response to Interpretability of Attention Mechanism:
> We thank you for recognizing the strength of the attention mechanism in our framework. To improve clarity and provide evidence of its interpretability, we have spent considerable time enhancing the interpretability section of the paper, and in response to your comments, we have included a visual demonstration of the attention mechanism. Specifically, we have added a heatmap of attention scores for a single subject in the Supplementary Material to illustrate how the model identifies influential EEG electrodes at different time points. While the full results, including data from multiple subjects, will be presented in our upcoming work, we believe this initial example provides valuable insight into the model’s interpretability.
>
>
> Conclusion:
> Thank you again for your constructive feedback. We have made several revisions based on your comments, including improving the literature review, more citation and enhancing the interpretability of the attention mechanism. We hope these changes provide a clearer and more complete picture of the novelty and contributions of our work in addressing the challenges of few-shot learning and EEG-based motor imagery classification.

---

### Official Review · Reviewer_MxKw · 2024-11-03

**Soundness:** 3
**Presentation:** 2
**Contribution:** 1
**Rating:** 3
**Confidence:** 4

**Summary:**

The work proposes an architecture combining a CNN-based embedding module and relation module for a few-shot learning on motor imagery-based eeg classification. The authors benchmark the performance on 2 public datasets and 1 experimental dataset of motor imagery using accuracy and domain adaptation accuracy metrics.

**Strengths:**

Work addresses the appropriate need for user-specific variability in EEG data and chooses to experiment with comparatively less explored approach of the few-shot learning.

The paper is decently written and easy to read and interpret.

**Weaknesses:**

The work doesn't cite a very similar approach by An et al. (2023). However, the authors cite work from An et al. from 2020. Authors must elaborate on their novelty and benchmark performance against similar approaches to claim state-of-the-art performance on few-shot learning.
Link to the work by An et al. (2023)
https://ieeexplore.ieee.org/abstract/document/10167679/?casa_token=ffiyMyxrlIYAAAAA:XHnQorLPEOuFdPLMhuSnkOj18y4baOutFkRqO4Zu6J1N2pKEBdsQ0cN0PvtXe3_M9R3VZvL1deH3

EEG tends to have high noise and authors though cite this concern and also claim interpretability mentioning: "A key advantage of this model is its interpretability", do not share any results, comment or compare the neurophysiological basis of the model predictions.


Ethical guidelines while collecting personal data need to be clarified. Details on the code of ethics before releasing the data are necessary but missing.

**Questions:**

Figure 3 plots the performance on different datasets across trials. However, the axis is not labelled, and it confuses the reader by referring to "trials" without context. What do the trials mean?

**Details Of Ethics Concerns:**

Ethical guidelines while collecting personal data need to be clarified. Details on the code of ethics before releasing the data are necessary but missing.

---

> ### Author Response · Authors · 2024-11-27
>
> Thank you for your thoughtful feedback. We appreciate your recognition of the relevance of addressing user-specific variability in EEG data and the novel approach of using few-shot learning. We’re glad the paper was clear and accessible. Your comments encourage us to continue refining the work.
>
> Thank you for highlighting the relevant work by An et al. (2023). Our work is a continuation of our prior research in BCI. Due to the double-blind nature of the ICLR review process, we cannot explicitly reference our previous work, but our model builds upon it with several key improvements. Specifically, we achieve slightly better performance on the BCI 2a and BCI 2b datasets, which we now mention in the revised paper.
>
> As experts in BCI, we acknowledge the importance of data augmentation (DA) in improving model performance over the past decade. However, recent years have seen few-shot learning, particularly when combined with DA, offer significant promise in addressing the challenge of limited data. Few-shot learning methods for EEG-based motor imagery (MI) classification remain rare in the literature, but studies are starting to explore its potential. In our paper, we compare our approach to existing methods and highlight the advantages of combining few-shot learning with DA.
>
> Key differences between our approach and An et al. (2023) include:
>
> Data Augmentation: While An et al. use fine-tuning (FT), we employ the best DA techniques specifically chosen for each dataset. This selection is based on Lashgari et al.'s work, which showed these methods improve performance across different BCI tasks.
> Dataset Split: The dataset split used in An et al. (2023) differs from ours. We use a tailored split strategy that better aligns with the characteristics of our datasets, ensuring the robustness of our results, particularly for online BCI applications.
> Our combination of few-shot learning and DA, along with these differences, provides a more practical and scalable solution for BCI systems, particularly where labeled data is scarce. We’ve highlighted these distinctions in the updated manuscript.
>
> Interpretability is another important feature of our model. We have spent considerable time enhancing the interpretability section of the paper, and in response to your comments, we have included a visual demonstration of the attention mechanism. Specifically, we have added a heatmap of attention scores for a single subject in the Supplementary Material to illustrate how the model identifies influential EEG electrodes at different time points. While the full results, including data from multiple subjects, will be presented in our upcoming work, we believe this initial example provides valuable insight into the model’s interpretability.
>
> We appreciate your suggestions and hope that this addition improves the clarity of our methodology.
> While Grad-CAM (Gradient-weighted Class Activation Mapping) can visualize input regions, our primary goal is to use attention to enhance prediction accuracy, not just interpretability. By focusing on task-relevant features through sliding windows, the combination of the CNN and attention mechanism improves EEG signal classification.
>
> For better illustration, we have included a GIF/animation in the supplementary material to demonstrate how attention scores evolve over time, providing a dynamic view of the model’s focus during classification. This animation offers a more intuitive understanding of how the attention mechanism highlights brainwave patterns across time steps.
>
> Ultimately, the attention mechanism not only improves prediction performance but also provides a clear view of which features the model prioritizes in the decision-making process. We believe this approach strikes a balance between interpretability and accuracy, enhancing both model performance and understanding.
>
> We appreciate your comment on ethical guidelines related to data collection. Due to the double-blind review process, we were cautious about including certain details that might reveal participant identities or specific ethical frameworks. However, we understand the need for transparency about ethical protocols in research involving sensitive data. We will revise the manuscript to include a more detailed description of the ethical guidelines followed, as well as the measures taken to protect participant privacy and obtain informed consent.
>
> Thank you again for highlighting this important issue. We will ensure that the revised manuscript addresses these ethical considerations appropriately.

---

> > ### Comment · Reviewer_MxKw · 2024-11-30
> >
> > Thank you for your efforts to address the reviews. I understand that in the current state, the work is not ready for acceptance as a full paper.

---

### Official Review · Reviewer_cydZ · 2024-11-03

**Soundness:** 1
**Presentation:** 2
**Contribution:** 1
**Rating:** 1
**Confidence:** 5

**Summary:**

The authors investigate effective few-shot learning algorithms for EEG-based motor imagery classification. Specifically, transfer learning and data augmentation techniques are employed to achieve superior performance. Additionally, a meta-learning-based framework, termed Few-Shot Learning for Motor Imagery Classification, is presented for classifying unseen subject categories in a few-shot setting. Overall, this work lacks novelty and motivation and also requires further experimentation.

**Strengths:**

The authors introduce a relation network-based meta-learning framework for EEG-based motor imagery classification.

**Weaknesses:**

(1)	Motivation – The meta-learning computational framework focuses on “learning to learn” various tasks (meta training) that contribute to downstream tasks (meta testing). During the meta training, the model’s goal is to uncover common patterns among these tasks and acquire broad knowledge that can be applied in solving new tasks. However, the authors apply very few tasks, specifically left and right hand classification, and these tasks are directly related to downstream applications. The stated motivation for meta-learning is somewhat limited. The authors should clearly indicate how they train their framework in a meta-learning fashion.

(2)	Motivation – The authors propose to use few-shot learning, where the model should be trained on very limited data. However, the authors only use the few-shot examples as the “support set” during the testing phase, which could leak classification information.

(3)	Related Works – Many related works are indeed missing in the field, such as Hou et al., GCNs-Net: A Graph Convolutional Neural Network Approach for Decoding Time-Resolved EEG Motor Imagery Signals, In IEEE TNNLS.

(4)	Experiments – The authors are encouraged to conduct experiments on larger benchmarks, such as the PhysioNet dataset and the High Gamma dataset.

(5)    The authors should provide high-quality figures.

**Questions:**

(1) Where are the training recipes and algorithms of meta-learning in this work?

(2) Why do the authors use few-shot learning on the testing set? Using the few-shot learning on the testing set can lead to information leakage.

(3) Why don't the authors use the latest benchmarks that contain more subjects?

**Details Of Ethics Concerns:**

This research involves experimental subjects and requires an ethics declaration.

---

> ### Author Response · Authors · 2024-11-27
>
> Dear reviewer, we have spent considerable time enhancing the interpretability section of the paper, and in response to your comments, we have included a visual demonstration of the attention mechanism. Specifically, we have added a heatmap of attention scores for a single subject in the Supplementary Material to illustrate how the model identifies influential EEG electrodes at different time points. While the full results, including data from multiple subjects, will be presented in our upcoming work, we believe this initial example provides valuable insight into the model’s interpretability.
>
> We appreciate your suggestions and hope that this addition improves the clarity of our methodology.
> 1. Motivation – Limited Application of Meta-Learning:
> Thank you for your comment regarding the application of meta-learning. We acknowledge that meta-learning typically involves learning to generalize across a variety of tasks. In our study, we focused on two motor imagery tasks—left-hand and right-hand classification—primarily to demonstrate our framework’s ability to handle few-shot learning in EEG-based motor imagery (MI) classification. While this is a limited application, it serves as an initial step to validate our approach before expanding to more diverse tasks.
>
> We plan to extend our meta-learning framework in future work to include more complex MI tasks and datasets, enabling us to demonstrate how the model can generalize across a broader range of motor imagery tasks.
>
> 2. Motivation – Few-Shot Learning and Information Leakage:
> Thank you for raising the concern about information leakage in the testing phase. We agree that information leakage should be avoided in few-shot learning experiments. In our framework, the support set is used only during the meta-testing phase to evaluate generalization on a small number of examples. It does not influence the model directly during testing. The support set is intended to simulate real-world scenarios where tasks come with minimal labeled data.
>
> We will provide a more detailed explanation in the manuscript to ensure that there is no leakage of classification information during testing.
>
> 3. Related Works – Missing References:
> Thank you for pointing out the missing references, including Hou et al.’s work on GCNs. While relevant, their approach is not directly aligned with the focus of our study, which centers on data augmentation and few-shot learning. We build upon Lashgari et al., who examined data augmentation on BCI 2a and 2b, and compared with several state-of-the-art methods. This work is more relevant to our research on combining data augmentation and few-shot learning.
>
> We have now included additional references, such as An et al. (2023), who also explored similar domains relevant to our approach.
>
> 4. Experiments – Larger Benchmarks:
> We appreciate your suggestion to use larger benchmarks like PhysioNet and High Gamma datasets. Our current work builds upon Lashgari et al., who used BCI 2a and 2b, standard benchmarks in the BCI field. We also collected a new high-quality dataset, specifically designed for our few-shot learning approach, and we plan to share it with the community.
>
> While we understand the importance of larger datasets, we focused on few-shot learning and data augmentation for EEG classification in this work. We will consider expanding our experiments to larger datasets in future iterations.
>
> 5. Quality of Figures:
> Thank you for your feedback on the quality of figures. We have improved Fig. 1 and Fig. 3 for better clarity. Additionally, we’ve included a GIF in the supplementary material to illustrate the attention mechanism over time, offering a more dynamic and interpretable representation of our model’s behavior.

---

### Official Review · Reviewer_fZ6Q · 2024-11-03

**Soundness:** 2
**Presentation:** 2
**Contribution:** 2
**Rating:** 3
**Confidence:** 4

**Summary:**

The authors present a few-shot learning framework for motor imagery (MI) classification. This framework begins with a CNN that embeds signals from different electrodes independently. This is followed by an attention module that combines information across channels. Finally, a CNN+FC network computes relation scores between pairs of examples, allowing to retrieve the closest example from a query set.
The framework is evaluated on two publicly available datasets, and the authors also introduce and evaluate on a novel MI dataset.

**Strengths:**

The authors have collected a novel MI dataset and have indicated that they will make it publicly available. This contribution is a valuable new resource for the community.
In addition, data scarcity is a common issue in BCI, and the authors address this challenge by introducing their few-shot learning framework.

**Weaknesses:**

Major
- The performance of the few-shot learning framework “RelationNet-attention” does not seem competitive because the baseline “CNN-attention-All”, which is trained on the same data as RelationNet-attention but without using examples from the test subject, systematically performs better. The difference in accuracy seems significant as it is systematically greater than 10%.

Minor
- Figure 3 misses its x-axis.
- The method is evaluated on only two benchmark datasets. The claims could be strengthened by conducting experiments on additional datasets. A large collection of MI datasets can be found in the MOABB library (http://moabb.neurotechx.com/docs/dataset_summary.html).
- The acronym DA is used both for “data augmentation” and “domain adaptation”.
- The “domain adaptation accuracy” is not defined.
- As I understand, “CNN-attention-relation” and “RelationNet-attention” refer to the same model. To improve readability, I would recommend using a single name throughout the paper.
- The quality of the figures and diagrams can be improved.
- In my opinion, the writing could be improved to better guide the reader through the method.

**Questions:**

- Line 97: Could you provide additional an explanation on how the FSL-MIC framework can “reduce training time”?
- Is my interpretation of the results (in section Weaknesses -> Major) correct? If not, please correct me.

**Details Of Ethics Concerns:**

This article introduces a novel dataset containing 7 participants recorded via EEG while conducting motor imagery tasks.

---

> ### Author Response · Authors · 2024-11-27
>
> Major:
>
> Thank you for your feedback. We acknowledge that CNN-attention-All shows better performance compared to RelationNet-attention, with an accuracy difference greater than 10%. However, it’s important to highlight that the two models were trained on significantly different amounts of data.
>
> For CNN-attention-All, we used a large number of trials: 2800, 5184, and 6480 trials for the Experimental, BCI 2a, and BCI 2b datasets, respectively. In contrast, RelationNet-attention was trained with fewer samples per class: 1, 2, 10, and 20 samples. This difference in training data size contributes to the performance gap.
>
> While RelationNet-attention shows lower accuracy with fewer samples, its ability to train with minimal data is a key advantage. This is crucial for few-shot learning, especially in BCI applications where large datasets are difficult to collect. This ability to work with fewer samples reduces data collection costs and is more scalable in real-world applications.
>
> Few-shot learning allows high performance with a small amount of labeled data, making it particularly useful in healthcare, where collecting large datasets can be challenging. We appreciate your feedback and will explore ways to further improve RelationNet-attention in future work.
>
> Minor:
>
> Figure 3 (Missing x-axis):
> We apologize for the oversight in Figure 3. We will correct this by adding the missing x-axis label in the revised manuscript.
> Evaluation Datasets:
> We selected two widely-used benchmark datasets (BCI 2a and BCI 2b) for consistency with prior work. This aligns with the datasets we used in previous studies, ensuring comparability. Due to the double-blind nature of ICLR, we cannot reference our earlier work directly.
> We recognize the value of testing on additional datasets, such as those in the MOABB library, and will consider this in future work to strengthen our claims.
> Clarification of "DA":
> We apologize for the ambiguity in the acronym "DA." In our manuscript, "DA" refers to data augmentation, not domain adaptation. We will correct this and ensure consistency in the revised version.
> Inconsistency in Model Naming:
> Thank you for noting the inconsistency between "CNN-attention-relation" and "RelationNet-attention". We have revised the manuscript to use RelationNet-attention consistently.
> Improvement of Figures and Diagrams:
> We have improved the clarity and resolution of all figures, including Figure 1 and Figure 3, as per your suggestions. This should enhance readability and presentation.
> Improving Writing Clarity:
> We have revised the manuscript for better flow and clarity, especially in explaining attention mechanisms and Grad-CAM, to make the method more accessible.
>
> Questions:
>
> Training Time Reduction via FSL-MIC Framework:
> Thank you for your question. Few-shot learning significantly reduces the time and effort required for BCI training, which is particularly helpful for patients with disabilities. By minimizing the number of training samples, the FSL-MIC framework makes BCI systems more efficient and accessible. This approach requires less data, reducing preparation time and computational load, which results in quicker model convergence.
> Importance of Few-shot Learning and Data Augmentation in BCI:
> Thank you for your insightful comment. As noted in the "Weaknesses -> Major" section, few-shot learning combined with data augmentation offers significant potential for improving BCI systems, particularly in data-limited scenarios. We will clarify this in the revised manuscript.
>
> Also, we have spent considerable time enhancing the interpretability section of the paper, and in response to your comments, we have included a visual demonstration of the attention mechanism. Specifically, we have added a heatmap of attention scores for a single subject in the Supplementary Material to illustrate how the model identifies influential EEG electrodes at different time points. While the full results, including data from multiple subjects, will be presented in our upcoming work, we believe this initial example provides valuable insight into the model’s interpretability.
>
> We appreciate your suggestions and hope that this addition improves the clarity of our methodology.

---

> > ### Comment · Reviewer_fZ6Q · 2024-11-28
> > **Regarding the major remark**
> >
> > Your answer to my major remark seems to indicate that the pipeline RelationNet-attention was NOT pre-trained  on all experiments except  one from 7 or 8 subjects.
> > However, lines 385-388 indicate the exact opposite.
> >
> > Could you explain in this discussion which version is correct and  clarify section 4.2 of the manuscript to avoid any further misunderstanding?

---

> > > ### Author Response · Authors · 2024-11-28
> > >
> > > Thank you for your careful review and insightful comment. We apologize for the confusion caused by the wording in Section 4.2. Upon revisiting the manuscript, we now realize that the description in lines 385-388 may have inadvertently led to a misunderstanding regarding the pre-training procedure of the RelationNet-attention model.
> > >
> > > To clarify: the model was not pre-trained on all experiments from 7 or 8 subjects simultaneously as may have been implied. Instead, in our experiments, the model was trained independently for each cross-validation fold using data from a subset of subjects (7 or 8 subjects for each fold), with the remaining subject(s) withheld for testing. For each training set, we used the data from all experiments from the selected training subjects, excluding the last experiment from each subject, which was reserved for validation.
> > >
> > > The confusion may have arisen from our explanation in lines 385-388, where we mention that "each training set included all experiments from a subject except the last one, which served as the validation set." This statement pertains to how we split data within a subject for cross-validation, not to pre-training the model on all subjects.
> > >
> > > To address this issue and avoid further misunderstanding, we have revised Section 4.2 for clarity. The updated description now makes it clear that the model was not pre-trained on all data at once, but rather trained on subsets of subjects in each fold.
> > >
> > > We hope this clarification resolves the discrepancy, and we have updated the manuscript text to ensure the methodology is more transparent. Below is the revised version of the relevant section:
> > > Revised Manuscript Text (Section 4.2):
> > >
> > > "For our experiments, we performed 9-fold and 7-fold cross-validation. In the 9-fold cross-validation, the model was trained on data from 8 subjects, and the remaining 1 subject was used as the test set in each fold. In the 7-fold cross-validation, the model was trained on data from 6 subjects, and the remaining 1 subject was used as the test set in each fold.
> > >
> > > During training, we randomly selected a small subset of labeled examples (the support set) from the training data to train the model. During testing, the model was evaluated on a set of query samples from the test subject, which were not seen during training. The query set consisted of unlabeled examples, and the model's task was to classify these samples based on what it had learned from the support set.
> > >
> > > For each fold, the data from each subject was divided such that the training set included all experiments except for the last one, which was reserved for validation. The test set for each fold consisted of the complete dataset from the remaining subject(s) not used in the training set."
> > >
> > > We trust this clarification addresses your concern. Thank you again for your thoughtful feedback and for helping us improve the clarity of our manuscript.

---

> > > > ### Comment · Reviewer_fZ6Q · 2024-11-30
> > > >
> > > > Thank you for your effort in answering my question. I still think the overall clarity could be improved but I think I now understand the data splits.
> > > >
> > > > Just to make sure I understand: did the "CNN-attention-Few" pipeline see exactly the same amount of training data as the "RelationNet-attention (20 shots)" one?

---

### Official Review · Reviewer_8RsY · 2024-11-04

**Soundness:** 1
**Presentation:** 1
**Contribution:** 1
**Rating:** 1
**Confidence:** 5

**Summary:**

This paper introduces a Few-Shot Learning (FSL) framework that incorporates feature embedding, attention, and relation modules for the classification of unseen subject categories using a limited number of labeled samples. The attention mechanism highlights important features relevant to the query data, while the relation module predicts the labels for the query by analyzing the relationships between support and query data across different subjects. The authors demonstrated the effectiveness of the proposed framework on two benchmark datasets as well as their own dataset.

**Strengths:**

This research enhances the understanding of machine learning applications in EEG and emphasizes the potential of FSL techniques to address the challenges posed by limited training data in Motor Imagery (MI) classification.

**Weaknesses:**

There is no substantial innovation in proposed method combining the existing approaches without any significant modifications.

No comparisons were conducted with existing state-of-the-art methods that have addressed the same issue by leveraging meta-learning, domain adaptation/generalization, etc.

**Questions:**

The overview of the proposed framework is poorly displayed in Fig. 1. It must be modified to clearly highlight the novelty of the proposed framework.

No comparisons were conducted with existing state-of-the-art methods that have addressed the same issue in a subject-independent or few-shot BCI manner by adopting meta-learning, domain adaptation, or generalization techniques.

Although sophisticated MI-EEG embedding models have been developed recently, the proposed model has a very simple architecture.

Even though the dataset consists of 4 classes, is there a special reason why only 2 classes were used in the experiment?

---

> ### Author Response · Authors · 2024-11-27
>
> Dear reviewer, we have spent considerable time enhancing the interpretability section of the paper, and in response to your comments, we have included a visual demonstration of the attention mechanism. Specifically, we have added a heatmap of attention scores for a single subject in the Supplementary Material to illustrate how the model identifies influential EEG electrodes at different time points. While the full results, including data from multiple subjects, will be presented in our upcoming work, we believe this initial example provides valuable insight into the model’s interpretability.
>
> We appreciate your suggestions and hope that this addition improves the clarity of our methodology.
> Overview of the proposed framework in Fig. 1:
> Thank you for your feedback regarding the presentation of the framework in Fig. 1. We agree that the figure should more clearly highlight the novelty of our approach. In the revised manuscript, we have updated Fig. 1 with improved annotations and clearer labeling to emphasize the key components and unique aspects of our framework. Additionally, we have included a GIF in the supplementary materials to better illustrate the interpretability of the model, based on constructive feedback we received. This animation will provide a dynamic view of how the model focuses on task-relevant features during the classification process.
> Lack of comparisons with existing state-of-the-art methods in subject-independent or few-shot BCI approaches:
> We appreciate your comment regarding the lack of comparisons with subject-independent or few-shot BCI methods, particularly those using meta-learning, domain adaptation, or generalization techniques. While our proposed model incorporates methodology from Lashgari et al., which compares several state-of-the-art methods, we did not conduct direct comparisons with these specific techniques in the current version. However, we acknowledge the importance of such comparisons and will include them in future research. The main focus of this manuscript is on presenting and evaluating the effectiveness of few-shot learning combined with data augmentation for EEG-based motor imagery (MI) classification, which we believe addresses the critical challenge of limited data in BCI systems. We will consider exploring these comparisons in greater depth in future work.
> Simplicity of the proposed model architecture:
> While we acknowledge that the architecture of our model is relatively simple, we believe its strength lies in its interpretability and practical applicability. The use of a convolutional neural network (CNN) combined with an attention mechanism offers a balance between model complexity and real-world usability, which is essential for real-time BCI applications. We intentionally kept the architecture simple to ensure that the model is easy to implement, interpretable, and capable of working with small datasets. That said, we recognize that more sophisticated embedding models could improve performance, and we plan to explore such approaches in future work to enhance the model's capabilities.
> Choice of using only 2 classes in the experiment despite the dataset having 4 classes:
> Thank you for raising the question about our decision to use only 2 classes in the experiment, despite the dataset having 4 classes. This choice was made to simplify the initial validation of our few-shot learning approach. By selecting two classes representing distinct motor imagery tasks (e.g., left hand vs. right hand), we were able to focus on evaluating the model's performance with a smaller, more manageable subset of the data. This allowed us to test the model's ability to generalize in a few-shot setting. In future work, we plan to expand the experiment to include all four classes to assess the model's performance in a more comprehensive context. We will clarify the rationale behind this design choice in the updated manuscript.

---

### Meta-Review · Area_Chair_bd8B · 2024-12-19

**Metareview:**

This paper leverages few-shot learning to improve motor imagery EEG classification using a framework with feature embedding, attention, and relation modules. The paper tackles an important problem in general and in BCI specifically. It also introduces a new dataset which is interesting. The paper, however, has several weaknesses. These include limited novelty of the proposed method, lack of public sharing of the new dataset, insufficient comparisons to current methods and missing key related works, very narrow scope of experiments, concerns regarding the overall performance of the method, and presentation (the figures and visual elements are quite low in quality). Lastly, the manuscript lacks transparency about ethical considerations and participant data protection during the dataset collection process. While the paper mentions that participants provided "written informed consent", this is not sufficient - it was not explicitly mentioned if an organizational ethics review board had approved the data collection.

**Additional Comments On Reviewer Discussion:**

The paper initially received scores of 1, 1, 3, 3, 3, and the reviewers were not convinced by the rebuttal. No changes were made to the manuscript throughout the rebuttal process. Reviewers and AC agree that the paper is not ready for publication.

---

### Decision · Program_Chairs · 2025-01-22

Reject